# A functional theory of bistable perception based on dynamical circular inference

**Pantelis Leptourgos**[1]*, **Vincent Bouttier**[2,3], **Renaud Jardri**[2,3,4]*, **Sophie Denève**[2]

**1** Department of Psychiatry, Connecticut Mental Health Center, Yale University, New Haven, Connecticut, United States of America, **2** Laboratoire de Neurosciences Cognitives & Computationnelles, ENS, INSERM U-960, PSL Research University, Paris, France, **3** Univ Lille, INSERM U-1172, Lille Neuroscience & Cognition Centre, Plasticity & SubjectivitY (PSY) team, Lille, France, **4** CHU Lille, Fontan Hospital, CURE platform, Psychiatric Clinical Investigation Centre, Lille, France

* pantelis.leptourgos@yale.edu (PL); renaud.jardri@chru-lille.fr (RJ)

**Data Availability Statement:** The Matlab codes can be found here: github.com/VincentBt/dynamical_CI_bistable.

**Funding:** P.L. was supported by a PSL Research University PhD fellowship (https://www.psl.eu/en).

## Abstract

When we face ambiguous images, the brain cannot commit to a single percept; instead, it switches between mutually exclusive interpretations every few seconds, a phenomenon known as bistable perception. While neuromechanistic models, e.g., adapting neural populations with lateral inhibition, may account for the dynamics of bistability, a larger question remains unresolved: how this phenomenon informs us on generic perceptual processes in less artificial contexts. Here, we propose that bistable perception is due to our prior beliefs being reverberated in the cortical hierarchy and corrupting the sensory evidence, a phenomenon known as "circular inference". Such circularity could occur in a hierarchical brain where sensory responses trigger activity in higher-level areas but are also modulated by feedback projections from these same areas. We show that in the face of ambiguous sensory stimuli, circular inference can change the dynamics of the perceptual system and turn what should be an integrator of inputs into a bistable attractor switching between two highly trusted interpretations. The model captures various aspects of bistability, including Levelt's laws and the stabilizing effects of intermittent presentation of the stimulus. Since it is related to the generic perceptual inference and belief updating mechanisms, this approach can be used to predict the tendency of individuals to form aberrant beliefs from their bistable perception behavior. Overall, we suggest that feedforward/feedback information loops in hierarchical neural networks, a phenomenon that could lead to psychotic symptoms when overly strong, could also underlie perception in nonclinical populations.

## Author summary

In cases of high ambiguity, our perceptual system cannot commit to a single percept and switches between different interpretations, giving rise to bistable perception. In this paper we outline a computational model of bistability based on the notion of circular inference, i.e. a form of suboptimal hierarchical inference in which priors and / or sensory inputs are reverberated and over-counted. We suggest that descending loops (i.e. reverberated priors) transform our perceptual system from a simple accumulator of sensory inputs into a

V.B. was supported by a ANR-16-CE37-0015 PhD fellowship led by R.J (https://anr.fr/). S.D. was supported by an ERC consolidator grant "Predispike" (https://erc.europa.eu/) and by the James McDonnell Foundation award "Human Cognition" (https://www.jsmf.org/). This research was also supported by: ANR-17-EURE-0017 FrontCog and ANR-10-IDEX-0001-02 PSL grants (Département d' Etudes Cognitives of the Ecole Normale Supérieure). The funders had no role in study design, data collection and analysis, decision to publish, or preparation of the manuscript.

bistable attractor, that switches between two highly-trusted interpretations. Using analytical methods we derive the necessary conditions for bistable perception to occur. We show that our dynamical circular inference model is able to capture many features of bistability, such as Levelt's laws and the stabilizing effects of intermittent presentation of the stimulus. Finally we make novel predictions about the behavior of psychotic patients.

## Introduction

All perceptual systems have one fundamental goal: to interpret the surrounding environment based on unreliable sensory evidence. In most cases, this task is performed very accurately, and the correct interpretation is found. Sometimes, perceptual systems fail to detect any meaningful interpretation (e.g., when sensory evidence is too degraded) or converge to the wrong interpretation (e.g., visual illusions [1,2]). Finally, a third possibility occurs (mainly in lab conditions [3]) when ambiguity is high; the system detects more than one plausible interpretations but instead of committing to one interpretation, it switches every few seconds, a phenomenon known as *bistable perception* [4]. Despite ongoing scientific efforts, there has been no unanimous agreement either on the causes of bistability or on its functional role.

The dominant mechanistic view on bistable perception suggests that it results from the competition between different neuronal populations, each of them encoding a different interpretation of the sensory signal [5]. The two populations suppress each other via lateral inhibition, while some form of slow negative feedback (e.g., spike frequency adaptation or synaptic depression) acts on the dominant population, weakening the interpretation that is currently perceived [6–11]. Additionally, injected noise renders irregular switching and in some models, it can even be the driving force of oscillatory behavior [12–15]. Although these models have proven quite successful in describing different experimental observations (and linking them to the underlying neural mechanisms), they do not address functional considerations about bistable perception.

To overcome this issue, other groups suggested functional models of bistability, largely based on the idea that the brain is an inference machine and perception is equivalent to a probabilistic process (e.g., [16]; see also [17,18] for predictive coding, or [19–21] for sampling). However, some crucial questions remain largely unanswered from a purely normative perspective, namely, (1) why would a system form such strong percepts based on ambiguous sensory evidence, but only in some cases, and why do the percepts persist in such a way instead of switching rapidly, and (3) how the behavior of individuals in bistable perception tasks may predict their performance in other probabilistic inference tasks.

In the present paper, we address the problem of bistable perception by proposing a functional model with a well-defined interpretation in terms of generic neural processes. Based on previous experimental findings, we suggest that bistability could be a perceptual manifestation of *circular inference* (CI), a form of belief propagation in which priors and likelihoods are reverberated in the cortical hierarchy and consequently corrupted by each other [22,23]. More specifically, bistable perception could be imposed by the presence of "descending loops", where high-level beliefs are combined with sensory representations (through feedback connections), and subsequently reinforce themselves (through feedforward connections). This results in the perceptual system "seeing what it expects" instead of the truly ambiguous image [24]. Of note, previous work from our group linked CI with pathological brain function, as in the case of schizophrenia [25] but also to a smaller extent with physiological functioning [26].

In the following sections, we derive the dynamics of inference in the presence of ambiguous sensory stimuli and inference loops. The consequence of CI is to replace what is normally a slow temporal integration of unreliable sensory evidence with a bistable attractor switching between two highly trusted interpretations. We demonstrate that such a model can reproduce well-known qualitative aspects of bistability, including the four Levelt's laws and the stabilizing effect of intermittent presentation, while it also makes testable quantitative predictions (e.g., about the behavior of patients suffering from schizophrenia). Since circularity arises from an imbalance between neural excitation and inhibition in recurrent brain circuits [24,27], our approach bridges normative interpretations of bistable perception with plausible underlying neural mechanisms.

## Methods

Here, we introduce a CI model of bistable perception and highlight its underlying functional assumptions. For reasons of clarity, we refer to the example of the Necker cube, an ambiguous 2D figure which is compatible with 2 different 3D cubes and generates bistability: a cube that is "*seen from above*" (later called the SFA interpretation) and a cube that is "*seen from below*" (later called the SFB interpretation) (Fig 1A). Note that the model can be generalized to any other stimuli inducing perceptual rivalry.

### Generative model

Our model postulates that bistable perception is triggered by the same mechanisms and computations that underlie normal perception. There is accumulating evidence that the brain uses its cortical hierarchy to represent the causal structure of the world [28,29]. Brain circuits invert this "generative model" to find the most likely interpretation of the noisy sensory information. In other words, perception can be viewed as an instance of hierarchical Bayesian inference [28,30] (Fig 1A). A particularly striking example of this inferential process is 3D vision (such as the perception of the Necker cube). The brain has no direct access to the 3D structure of the perceived object. In contrast, it receives low-level 2D sensory information from the retina. In such a context, the task of the perceptual system is to extract valuable depth cues and combine them with high-level prior knowledge, to make "educated guesses" about the 3D object. Evidence suggests that this is a gradual process [31], with different brain regions representing features of different complexity; the lower levels of the visual cortex represent the basic features of the stimulus such as contours and orientations while higher levels are responsible for more abstract information such as the 3D organization of the stimulus [32,33].

In the case of the Necker cube, a veridical percept would correspond to a 2D drawing of crossing lines. The presence of illusory depth cues forces the brain to consider a 3D structure. Nonetheless, since the cues are ambiguous and contradictory, the 2D projection of the hypothetical 3D stimulus is compatible with different objects, including the SFA and SFB interpretations mentioned previously. The two interpretations are considered mutually exclusive, an assumption that corresponds with the epistemological truth that two different 3D objects cannot occupy the same space [17]. It is interesting to note that in a more general sense, the Necker cube is compatible with an infinity of 3D objects, among which the brain represents only the two symmetrical cubes. This reduction of possible causes could be the result of hyperpriors used by the brain and is not considered in the current model.

We formalize this inference problem with a simple graphical model, a chain with 2 latent variables and one sensory observation (Fig 1A). This "generative model" summarizes assumptions made by our sensory system on the underlying causes of natural inputs, which may significantly differ from the artificial data presented in a laboratory setting.

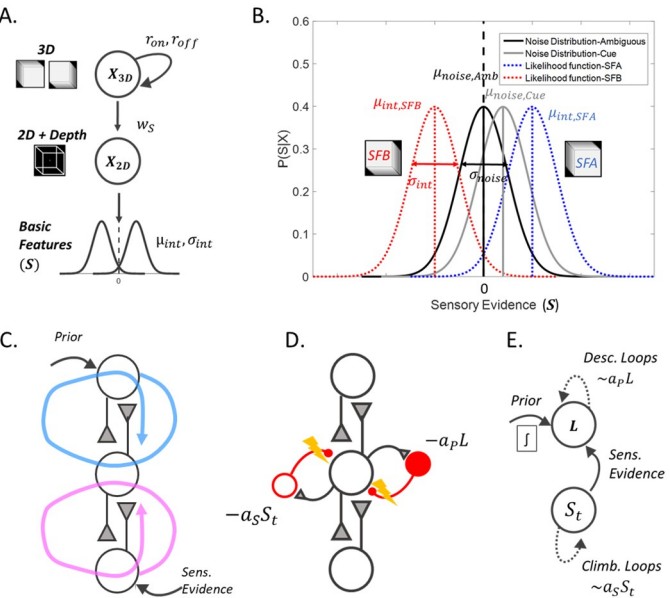

**Fig 1. Normative model for how 3D objects result in particular sensory inputs, and putative neural implementation of the corresponding perceptual inference.** (A.) The internal model is a simple Bayesian generative model, where 3D objects predict the 2D image, and the 2D image predicts low-level sensory inputs. The brain interprets the depth cues (basic features) as indicative of real depth. Consequently, it first reconstructs the 2D figure and from that, it infers the 3D object. Note that in reality there is one single 2D stimulus (the Necker cube drawing) containing contradictory depth cues. (B.) Close-up on the assumed "basic feature" distributions (likelihood) compared to the real input distributions. The brain interprets the depth cues as meaningful, predicting separate input distributions for the two cubes (SFA, SFB; two objects cannot occupy the same space), which corresponds to two nonoverlapping likelihood distributions in the internal model (dotted red and blue distributions). In the totally ambiguous case (cube with no extra cues), the real input is sampled from a distribution with mean 0 (black). Visual cues shift this input distribution toward mostly positive or negative values. Crucially, there is a discrepancy between the real input and the input assumed by the internal model. This, together with the loops, predicts the suboptimal inference at the heart of bistable perception. (C.) A simplified neural implementation of hierarchical perceptual inference. Reciprocal connections can combine bottom-up sensory evidence with top-down priors at all levels of the hierarchical representation. Unfortunately, this also creates redundant information loops, ascending (magenta arrow) and descending (blue arrow). (D.) The brain can cancel these loops by using inhibitory interneurons and maintaining a tight E/I balance. If this balance is impaired, however, there will be some residual loops, parameterized by $a_P$ (descending loops, amplifying prior beliefs) and $a_S$ (ascending loops, amplifying the sensory evidence). L is the log-ratio of the belief. (E.) From the Bayesian model (A.) we derived an attractor model that performs inference in the presence of loops. The model accumulates noisy evidence while descending loops add positive feedback and ascending loops increase the sensory gain.

The sensory observation ($S$) represents the basic features extracted by visual receptors (edges, contrast, etc.). For simplicity, $S$ is assumed to be a scalar drawn from two probability distributions, one for each configuration of the cube, as illustrated in Fig 1A and 1B (red and blue dotted distributions; $P(S|X_{2D} = 1) \neq P(S|X_{2D} = 0)$). These distributions have different means $\pm\mu_{int}$ and the same variance $\sigma_{int}^2$. The difference in these two distributions considers the fact that natural 3D objects have true depth cues (disparities, shadows, occlusion, etc.), predicting different likelihoods for the two interpretations. Note that completely ambiguous stimuli (i.e., falling in the perfect overlap between the two distributions) are, in fact, rarely encountered in nature.

The next variable $X_{2D}$ is binary and represents an intermediate level of complexity in the perceptual hierarchy (e.g., the 2D surfaces and their orientation). Finally, the binary variable $X_{3D}$ represents the final 3D cube configuration, with values 0 and 1 corresponding to SFB and

SFA respectively. $w_S$ corresponds to how reliably $X_{3D}$ predicts $X_{2D}$.

$$w_S = P(X_{3D} = 1 \mid X_{2D} = 1) = P(X_{3D} = 0 \mid X_{2D} = 0) \tag{1}$$

We also assume that the environment has some volatility, e.g., objects are not permanently present, but occasionally appear or disappear. Thus, $X_{3D}$ can randomly switch at any time, as represented by two rates of change, from 0 to 1 ($r_{on}$), and from 1 to 0 ($r_{off}$). For the sake of simplicity in the notation, we will replace $X_{3D}$ at time t by $X_t$, representing the 3D configuration of the cube (SFA or SFB) at time t.

$$r_{on}dt = P(X_t = 1 \mid X_{t-dt} = 0) \tag{2}$$

$$r_{off}dt = P(X_t = 0 \mid X_{t-dt} = 1) \tag{3}$$

Note that if we use $r_{on} \neq r_{off}$, one of the two interpretations becomes more probable that the other. This is very useful in the case of the Necker cube, where people usually prefer the SFA interpretation, according to a general prior to view things from above ($r_{on} > r_{off}$) [34].

Now that we have described the generative model, i.e., the internal model used by the brain to perceive objects in the real world, we have to consider the artificial stimulus provided during a bistable perception experiment. The Necker cube is very unnatural in the sense that it contains no real depth cue. Thus, the sensory information it provides is assumed to be sampled (independently at each time step) from a Gaussian distribution with mean $\mu_{noise}$ ($\mu_{noise} = 0$ (Gaussian process without drift) if the cube is completely unbiased and $\mu_{noise} \neq 0$ (Gaussian process with drift), if there are visual cues supporting one of the two configurations, e.g., different contrast for the edges) and variance $\sigma^2_{noise}$ (Fig 1B; black and gray distributions).

The ultimate goal of the perceptual system is to infer $X_{3D}$ using the noisy measurements and any available prior knowledge (for more information about the generative model, see S1 Text).

## Temporal dynamics of inference

We show in S1 Text that exact inference implements a leaky integration of the noisy sensory input (Fig 1E), i.e.

$$\frac{dL}{dt} = -\Phi(L) + w_{int}S \tag{4}$$

where $w_{int} = \frac{2\mu_{int}}{\sigma^2_{int}}(2w_S - 1)$ represents the overall reliability of the sensory input (as assumed by the generative model). $L$ is the log-odds ($L = \log\left(\frac{P(X_t=1|S_{0\rightarrow t})}{P(X_t=0|S_{0\rightarrow t})}\right)$). The nonlinear leak term $\Phi(L)$ depends on the transition rates, i.e.,

$$-\Phi(L) = (r_{on}e^{-L} - r_{off}e^{L}) + (r_{on} - r_{off}) \tag{5}$$

As a result of this leak, in the absence of sensory evidence, the log-odds go back to the constant prior value $\log\left(\frac{r_{on}}{r_{off}}\right)$. This relaxation is faster for larger volatility in the environment (higher transition rates). In the presence of reliable and unambiguous sensory input (e.g., when adding visual cues, i.e., $\mu_{noise} \neq 0$), $L$ integrates out the noise and eventually reaches high (positive) or low (negative) values, corresponding to high levels of confidence in favor of the SFA or SFB configurations. However, in the presence of a completely ambiguous sensory input, $L$ integrates unbiased noise ($\mu_{noise} \neq 0$) and constantly hovers around the prior value, rarely reaching a sustained high level of confidence in either of the two configurations.

Dynamics notably change in the presence of CI. CI is defined in the context of hierarchical probabilistic inference but can also be understood intuitively as a consequence of feedforward/feedback loops in brain circuits (Fig 1C). Bottom-up sensory evidence (from $S$ to $X_{2D}$) and top-down prior information (from $X_{3D}$ to $X_{2D}$) have to be combined to compute the probability of intermediate representations ($X_{2D}$), a task presumably performed by feedforward (bottom-up) and feedback (top-down) connections converging on the same intermediate "2D" sensory area [35]. This hypothesis is supported by the experimentally observed top-down modulation of sensory neuron responses by higher-level interpretation of the image [36–38]. However, feedforward connections between the "2D" and "3D" areas also communicate this modulated sensory response back to the "3D" areas. While this modulation does not bring any "new" information, it could nevertheless be mistaken for additional sensory evidence supporting the current interpretation. In fact, without dedicated control mechanisms, feedforward/feedback loops would systematically result in CI in the underlying perceptual process. We found previously that while this can, in theory, be avoided by maintaining a tight excitatory/inhibitory balance in brain circuits (Fig 1D), human subjects show some level of circularity in their probabilistic reasoning, which is aggravated in individuals suffering from schizophrenia [25,26].

Here, we quantify the strength of CI by two variables representing the level of "ascending" (also called "climbing" [22]; $a_S$) and "descending" loops ($a_P$). Descending loops represent to what extent top-down modulation of sensory responses is misinterpreted by upstream (higher-level) neurons as new sensory information, forcing the perceptual system to "see what it expects". Vice-versa, ascending loops represent to what extent intermediate sensory responses are misinterpreted by downstream (lower-level) neurons as prior knowledge, even when they do not provide them with any new information (Fig 1C). This forces the perceptual system to "expect what it sees" and over-interpret weak sensory inputs.

If CI is introduced in the model, the dynamics of perceptual integration changes as follows (Fig 1E):

$$\frac{dL}{dt} = -\Phi(L) + aL + w_{int}^* S \tag{6}$$

Note that the new auto-amplification term $aL = 2a_P w_S L$ (due to the corruption of the sensory evidence by the prior belief) is proportional to the strength of descending loops $a_P$ and the assumed reliability of the sensory information, $w_S$. If $a$ is large enough, this amplification term may exceed the leak term, at least in a certain range of confidence near $L = 0$. This leads, as we will see, to bistable dynamics. Importantly, this term not only depends on the strength of the descending loops but also on the reliability of the sensory input (assumed by the generative model). Bistable dynamics occur only for large $w_S$, which we may interpret as a typically highly reliable input (such as 2D drawings of 3D objects) as opposed to typically unreliable inputs (e.g., low contrast or degraded stimuli). This may explain in part why bistable perception is a relatively rare phenomenon in natural (nonlaboratory) settings.

In contrast, ascending loops amplify the weight of the sensory evidence according to their strength, i.e., $w_{int}^* = w_{int}(1 + 2w_S a_S)$. In particular, ascending loops affect the dynamics only if a sensory stimulus is present and tend to destabilize the percept by increasing the gain of the noise injected into the dynamical system.

Note that without loss of generality, this model of perceptual dynamics can be reduced to 4 free parameters: the two transition rates $r_{on}$ and $r_{off}$, the auto-amplification $a$ and the overall gain of the sensory inputs $w_{int}^*$.

## Perceptual decision

Finally, we require a model of perceptual decision, which can predict the current percept from the confidence. For simplicity, we assume a maximum-a-posteriori (MAP) decision criterion, which means that decisions are made according to the sign of $L$ (SFA if $L > 0$; SFB if $L < 0$). The MAP decision criterion results in optimal behavior when the goal of the system is to maximize accuracy, as in the case of perception.

## Simulations

For all the simulations, we used the Euler–Maruyama algorithm. The time step was fixed at $d_t = 0.01s$. Both the standard deviation of the noise $\sigma_{noise}$ (real model) and of the likelihood function $\sigma_{int}$ (internal model) were equal to 1. The mean of the likelihood function $\pm\mu_{int}$ was also fixed at $\pm 1$. $\mu_{noise} = 0$ for the completely ambiguous case and $\mu_{noise} \neq 0$ when sensory evidence was biased. The initial belief in all simulations was $L_0 = 0$. A summary of the parameters can be found in Table 1.

## Results

As a first step, we highlight the importance of the descending loops in the generation of bistable perception from a phenomenological and mechanistic point of view. Subsequently, we illustrate how CI replicates some of the most seminal features of bistable perception, such as Levelt's laws but also some counterintuitive findings, including stabilization of perception after a brief disappearance of the stimulus. Finally, we present further consequences of the model, notable predictions about the performance of schizophrenia patients exposed to bistable stimuli.

## Strong descending loops induce bistable perception

An example of model dynamics in response to a continuous presentation of a Necker cube, in the presence of strong descending loops is shown in Fig 2A and 2C.

With descending loops, the percept switches between two highly trusted interpretations (for example, $L = 4$ corresponds to probability 0.98 in favor of SFA; see also S3 Text). Periods with low confidence are short and limited to sudden perceptual switches, induced by the noisy input. These switches occur at apparently random times, resulting in an exponential decay

**Table 1. Parameters of the model.**

| Variable | Description | Link to other variables |
|---|---|---|
| $\mu_{noise}$ | Drift of sensory evidence | - |
| $\sigma_{noise}$ | Standard deviation of sensory evidence | - |
| $\mu_{int}$ | Mean of likelihood function | - |
| $\sigma_{int}$ | Standard deviation of likelihood function | - |
| $w_S$ | Feed-forward weight | - |
| $a_P$ | Descending loops | - |
| $a_S$ | Ascending Loops | - |
| $r_{on}$ | Transition rate (0➜1) | - |
| $r_{off}$ | Transition rate (1➜0) | - |
| $w_{int}$ | Sensory gain without ascending loops | $w_{int} = \frac{2\mu_{int}}{\sigma_{int}^2}(2w_S - 1)$ |
| $w_{int}^*$ | Sensory gain with ascending loops | $w_{int}^* = w_{int}(1 + 2w_S a_S)$ |
| $b$ | Bias | $b = r_{on} - r_{off}$ |

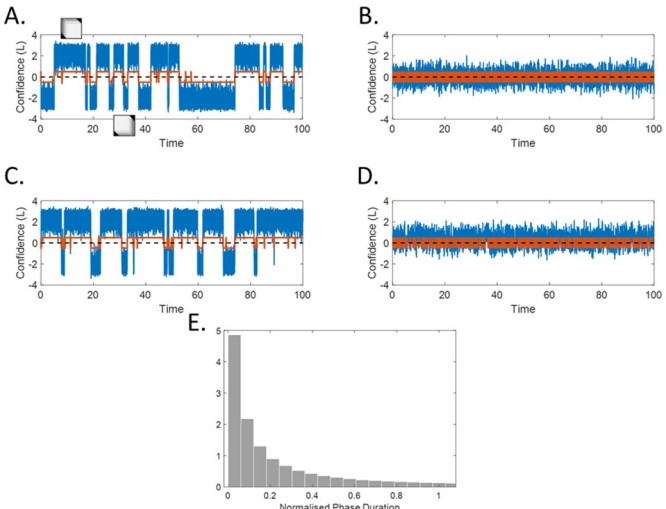

**Fig 2. Examples of model dynamics. (A.)** Model with descending loops ($a_P = 1.5$), unbiased ($r_{on} = r_{off} = 0.5$), with sensory gain $w_{int} = 0.8$. The model received an ongoing, ambiguous, white noise input with standard deviation $\sigma_{noise} = 1$. Blue line: L (log-ratio of the belief / confidence), red line = percept, dashed line = decision threshold). **(B.)** Model with no descending loops (same parameters as in (A.) except $a_P = 0$). **(C.)** The same model as (A.), but with a preference for the "SFA" configuration (transition rates changed to $r_{on} = 0.52$, $r_{off} = 0.48$). **(D.)** The same model as (B.), with $r_{on} = 0.6$, $r_{off} = 0.4$. **(E.)** Phase-duration histogram (No loops; unbiased). The dynamical circular inference model (with/without loops; with/without bias) predicts exponential distribution of phase-durations. Gamma-like distributions, often observed in bistable perception experiments, can be obtained by adding filtered noise, adaptation-like mechanisms or more complex decision criteria to the model (see Discussion).

observed in the distribution of dominance durations (Fig 2E). When there is a bias (e.g., $r_{on} > r_{off}$), one of the two configurations (e.g., SFA) becomes more likely and is perceived more often (Fig 2C). However, the shape of the dominance durations remains similar for the two configurations, even if the durations of the preferred configurations are longer overall. It's worth-highlighting that the stronger interpretation is also perceived with higher confidence, a prediction that could be tested in future studies.

For comparison, we also show the dynamics of the model without descending loops ($a_P = 0$) (Fig 2B and 2D). The resulting system is equivalent to a hidden Markov model (HMM), with transition rates $r_{on}$ and $r_{off}$ [39], and has only one stable state corresponding to the prior. As a result, the confidence behaves similarly to a leaky random walk. Since the leak maintains L close to zero, the system rarely attains high levels of trust in either configuration, which may preclude the emergence of strong and stable percepts in the absence of descending loops (instead, low confidence might give rise to mixed percepts [40]).

## Dependency of bistability on the parameters

Due to its simplicity, the model dynamics can be analyzed more formally. This has the advantage of generalizing the model and providing a general view on the dependency of bistable perception on prior assumptions about the external world and on the strength of ascending and descending loops.

This dynamics can be represented by an energy landscape plotting the "potential" (the temporal integral of the dynamic Eqs (1)/(6)) as a function of the current state L. The relationship between the energy landscape and stability of a dynamical system is shown in Fig 3A and 3B, while the actual energy landscape of the model for different parameter settings is shown on Fig 3C and 3D. In the absence of inputs, L always decreases toward the lower potentials in these

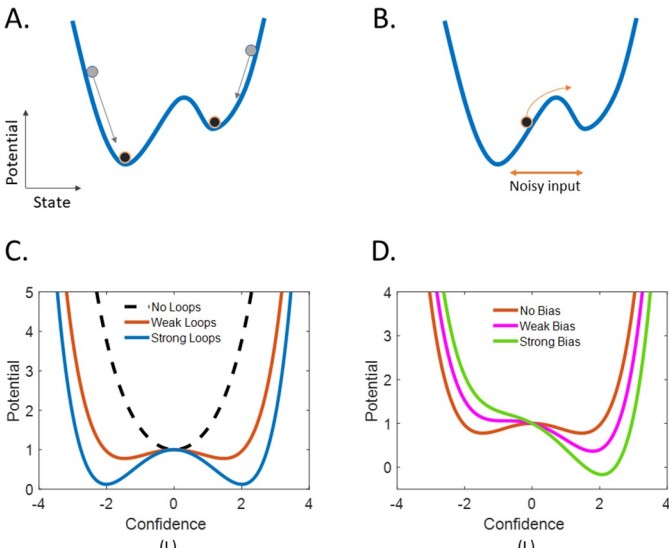

**Fig 3. Energy landscapes of the model with and without descending loops.** (**A.**) Schema illustrating the relationship between wells in the energy landscape (potential = integral of the dynamic equation, in blue) and stable states. Gray and black dots represent the initial and final state from two different initial states. In the absence of external input, dots can only decrease. (**B.**) Schema illustrating how noise can force the state to climb an energy barrier (a hill in the energy landscape) and switch to a different stable state. (**C.**) Energy landscape of the model with no descending loops (dashed, $a_P = 0$), and two increasing levels of descending loops (red: $a_P = 1$, blue: $a_P = 1.3$). Descending loops generate a bistable attractor, whose stable fixed points correspond to (strong beliefs about) the two interpretations (blue). In contrast, a system with no loops has only one attractor, the prior, (equal to 0 in this unbiased scenario). (**D.**) Energy landscape for different biases, no bias (red: $r_{on} = r_{off} = 0.5$), weak bias (magenta:, $r_{on} = 0.55$, $r_{off} = 0.45$) and strong bias (light green, $r_{on} = 0.6$, $r_{off} = 0.4$). Note that for stronger biases, the nonpreferred configuration becomes unstable.

energy landscapes, until it reaches a stable fixed point corresponding to a local minimum in the potential, also called an "energy well" (Fig 3A). The presence of a noisy input introduces random perturbations which might allow $L$ to temporarily climb the barrier between two wells, thus switching to a different stable state (Fig 3B).

Without the descending loops, the model is equivalent to an HMM. Importantly, an HMM acts as a leaky integrator with only one stable fixed point (the prior) determined by the 2 rates (volatility):

$$L_{St, a=0} = \log\left(\frac{r_{on}}{r_{off}}\right) \tag{7}$$

This can be visualized by observing that the corresponding energy landscape contains a single energy well (Fig 3C, dashed line). As long as the descending loops are weak compared to the leak, the prior remains the only fixed point of the system and is stable. For example, with $r_{on} = r_{off} = r$, this remains true up to the value:

$$a_P^{Pf} = \frac{r}{w_S} \tag{8}$$

At this value, the system undergoes a pitchfork bifurcation (Fig 4A; see also S2 Text). The preexisting fixed point becomes unstable and 2 additional attractors are generated, given by the 2 symmetrical, nonzero solutions of the equation $-\Phi(L) + aL = 0$ (Figs 3C and 4A). The stronger the descending loops (or the weaker the leak), the further apart the 2 symmetrical

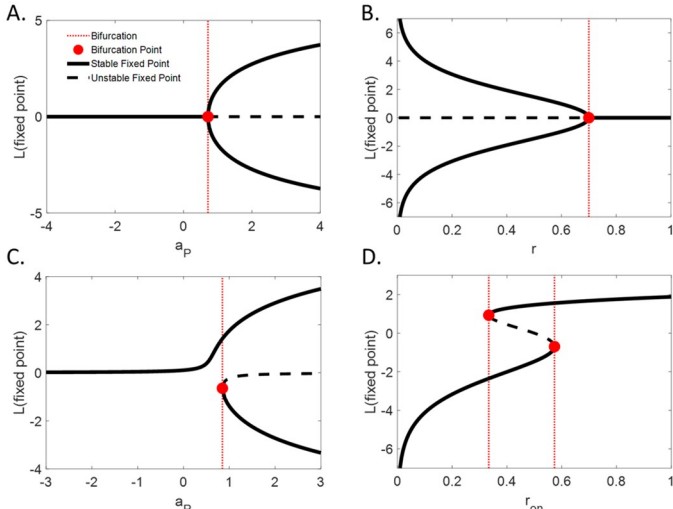

**Fig 4. Phase diagrams of the model dynamics.** **(A.)** Stable fixed point (plain), unstable fixed point (dashed) and bifurcation point (red dot) as a function of $a_P$ for an unbiased system ($r_{on} = r_{off} = r$). **(B.)** Stable fixed point, unstable fixed point and bifurcation points as a function of r. **(C.)** The same as (A.) for a biased system ($r_{on} > r_{off}$). **(D.)** The same as (B.) but as a function of $r_{on}$, $r_{off}$ being fixed at 0.5. Note that bistability can exist in a narrow range around symmetry. **(A.,B.)** Pitchfork bifurcation for symmetrical systems. **(C.,D.)** Saddle-node bifurcation for asymmetrical systems.

attractors are, resulting in more highly trusted configurations, which are also more stable since the energy barrier is harder to cross.

Adding bias to the system ($r_{on} \neq r_{off}$; e.g., SFA bias in Necker cube) creates an asymmetry in the energy landscape (Fig 3D). A saddle-node (SN) bifurcation occurs when the loops become strong enough to overcome the leak (Fig 4C; for a mathematical description of the SN bifurcation, see S2 Text). However, bistability can only exist in a narrow range of biases (i.e., the difference between the two transition rates $r_{on}$ and $r_{off}$), more particularly in the range constrained by the 2 SN bifurcation points (one for $r_{on} > r_{off}$ and one for $r_{on} < r_{off}$; Fig 4D). These two bifurcations represent points at which the bias becomes strong enough to ensure that only one of the two configurations (the most likely one a-priori) can be stably perceived.

Our analysis suggests that descending loops can constitute a crucial part of the machinery of a system exhibiting bistable perception. When they are strong enough to overcome the effect of the leak, they generate a bistable attractor, implementing a memory-like mechanism that pushes the belief toward more extreme values based on the previous observations. This helps the system make decisions and act upon them in the absence of fully convincing evidence.

Until now, our analysis focused mainly on the effects of the descending loops. However, ascending loops play an important role as well. According to (6), ascending loops increase the gain of the sensory evidence (noise) (Fig 3B), which consequently acts by destabilizing perception and reducing the effect of the bias on predominance.

In conclusion, this analysis demonstrates that robust bistable perception requires a very specific set of conditions. It can only exist if there is a combination of (1.) reliable sensory inputs (large $w_S$), (2.) stimuli that are assumed to be stable (i.e., small transition rates $r_{on}$ and $r_{off}$, that are dominated by descending loops), (3.) at least two probable interpretations, even if one can dominate the other (i.e., $r_{on}$ and $r_{off}$ relatively close to each other, leading to a weak bias). Given these stringent conditions, it is not surprising that bistability is rather uncommon in everyday life and occurs mainly for artificial stimuli chosen to obey these requirements.

In the next sections, we explore the predictions of the model regarding well-known psychophysical features of bistable perception.

## Levelt's laws

An important qualitative aspect of bistable perception is Levelt's laws. These laws constitute a set of 4 psychophysical propositions relating the strength of the bistable stimulus to the phenomenology of binocular rivalry [41], and more generally of bistable perception [42]. Despite some recent modifications in their formulation (to account for new experimental data [43,44]), Levelt's laws remain fundamental to our understanding of the machinery of bistability and an important crash-test for any potential model. We will present one by one the four revised propositions (as described in [42] and not in Levelt's original monograph [41]) and will critically discuss them through the prism of the *dynamical circular inference* (dCI) model.

**1st Levelt's law.** The first proposition links the stimulus strength with the predominance of each interpretation. It postulates that increasing the stimulus strength of one perceptual interpretation increases the predominance of this perceptual interpretation [42]. For example, adding a cue to the Necker cube helps the relevant interpretation gain more perceptual dominance compared to its rival. Although in modern terminology, proposition 1 sounds more like a tautology, it is still useful for detecting stimulus features (or parameters of the model) that affect the strength of an interpretation [44]. Within our model, we can parameterize the strength of the sensory evidence by adjusting the drift $\mu_{noise}$ of the Gaussian noise, which biases the sampling of evidence (Fig 1B). As expected, the more positive the drift the closer the relative predominance goes to 1 (the opposite for negative drift) (Fig 5A), in agreement with the first proposition.

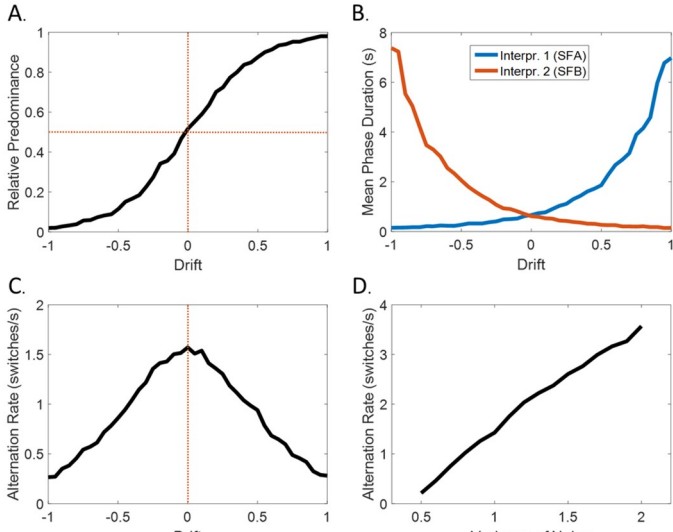

**Fig 5. Levelt's laws.** The circular inference model qualitatively reproduces the 4 Levelt's propositions (here: $w_S = 0.9$; $a_P = 1$; $r_{on} = r_{off} = 0.5$). **(A.)** 1st proposition—increasing the stimulus strength of one perceptual interpretation increases the predominance of this perceptual interpretation. **(B.)** 2nd proposition—Manipulating the stimulus strength of one perceptual interpretation of a bistable stimulus does not equally influence the average dominance duration of both interpretations, but mainly affects the persistence of the stronger interpretation. **(C.)** 3rd proposition—Increasing the difference in the stimulus strength between the 2 perceptual interpretations should result in a decrease in the perceptual alternation rate (i.e., maximum number of switches at equi-dominance). **(D.)** 4th proposition—When we increase the strength of both interpretations, the number of switches increases.

**2nd Levelt's law.**    The second proposition is less intuitive than the first and posits that manipulating the stimulus strength of one perceptual interpretation of a bistable stimulus does not influence equally the average dominance duration of both interpretations, but mainly affects the persistence of the stronger interpretation [42,45]. For example, increasing the strength of a visual cue in the Necker cube example mainly affects the mean dominance duration of the corresponding interpretation. The dCI model is fully compatible with Levelt's second law, as presented in Fig 5B; making the drift more positive (bias for SFA) predominantly affects the mean phase duration of the SFA interpretation (the opposite happens for a negative drift and the SFB interpretation). Indeed, the drift acts as an additional bias term in (4)/(6), which deepens the well of the strong interpretation, while making the other well shallower. This dual effect of the drift (not obvious in other models in which different variables represent the different interpretations, see also [12]), along with the model's inherent nonlinearity can explain Levelt's second law [45].

**3rd Levelt's law.**    Levelt's third proposition is closely related to the second proposition [44] and suggests that increasing the difference in the stimulus strength between the 2 perceptual interpretations should result in a decrease in the perceptual alternation rate [42]. In the Necker cube example, this proposition implies that adding a visual cue results in fewer switches. Importantly, the dCI model behaves exactly as the third proposition dictates. As shown in Fig 5C, the alternation rate achieves its maximum value for drift = 0 (completely ambiguous stimulus) and decreases symmetrically as the drift becomes more positive or negative, a direct consequence of the third law [45].

**4th Levelt's law.**    Finally, the fourth proposition goes one step further and discusses what happens to the alternation rate if we equally increase the strength of both interpretations. In this case, the number of switches increases, resulting in a higher alternation rate. Contrary to the 3 first propositions, the fourth proposition illustrates the effect of a simultaneous and equal manipulation of both interpretations (global stimulus strength). In the model, this should result in an increase in the mean of the absolute value of the sensory evidence, while it should have no effect on the mean of the sensory evidence per se. In other words, this global manipulation can be captured by a change in the variance in the noise distribution $\sigma_{noise}$. A higher variance results in more exploration of the energy landscape due to the noise. Consequently, as illustrated in Fig 5D, increasing $\sigma_{noise}$ results in more switches, in agreement with Levelt's fourth law.

In conclusion, the model obeys Levelt's laws regardless of the chosen parameters as long as

1. The sensory gain is high enough to induce transitions.

2. The bias is not strong enough to render one of the two configurations unstable.

Note that the respect of Levelt's laws is not sufficient to prove the presence of descending loops since the model without loops can also reproduce them (as long as the decision threshold is set appropriately). However, definite support for the existence of descending loops is provided by the stabilization of the percept by intermittent presentations of the stimulus, as described in the next section.

## Intermittent presentation

When an ambiguous stimulus is presented continuously, switches between competing interpretations occur randomly every few seconds, with consecutive phase durations being largely independent [46]. Based on this observation, many researchers concluded that bistable perception is principally a memoryless process ([47], see also [48,49]). Nevertheless, this conclusion contravenes another observation: the fact that people tend to perceive the same interpretation

repeatedly when ambiguous stimuli are presented intermittently for a wide range of OFF-durations (intervals during which stimulus is absent) [50,51]. This second observation forced researchers to assume the presence of some perceptual memory [52], which manifests when the stimulus disappears from the screen. A variety of mechanisms implementing this memory have been proposed, including low-level mechanisms such as adaptation (combined with sub-threshold effects; [9]), or high-level memory mechanisms located outside the extrastriate cortex [51,53,54]. The dCI model offers a different explanation for this stabilization effect, based on the descending loops.

In agreement with previously published experimental observations, our model predicts no significant correlation in the duration of successive phases [46,47], as expected from a model that does not contain adaptation (or adaptation-like) mechanisms [49]. However, the model should be able to predict a stabilization effect, when the stimulus disappears for brief durations. To quantify stabilization, many studies referred to the alternation rate, which is the number of switches in a time interval [50,51,55]. However, this measure is not ideal as it can be affected by various confounding factors including different presentation durations and switches occurring during ON-durations (interval during which stimulus is present). Moreover, the alternation rate considers both interpretations together and obscures any possible asymmetries. Instead, we used the survival probability (SP) of each interpretation, which is the probability that the dominant percept at the end of an ON-duration will be dominant again when the stimulus reappears after the OFF-duration. Fig 6A illustrates our interpretation of the phenomenon (5 ON-OFF cycles, $a_P > 0$).

Without descending loops ($a_P = 0$), and in the absence of input (i.e., when the stimulus is "OFF"), the belief progressively goes back to its prior value ($\log\left(\frac{r_{on}}{r_{off}}\right)$) due to the leak (Fig 6B

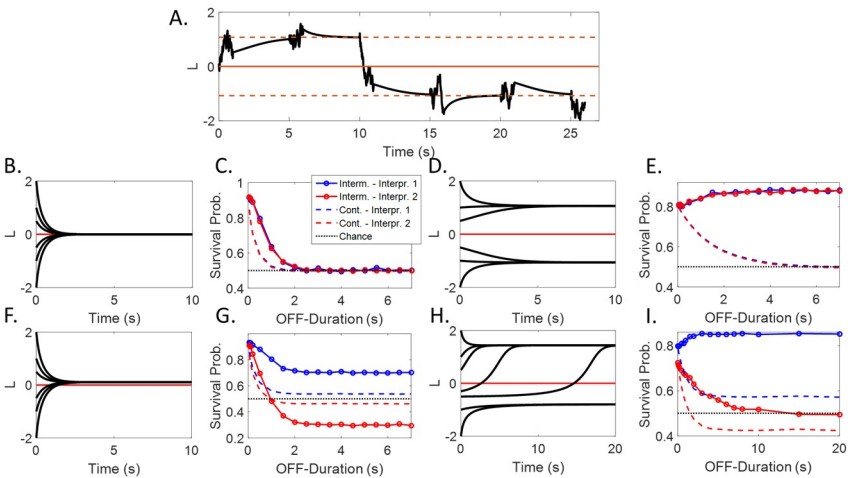

**Fig 6. Continuous vs intermittent presentation. (A.)** An interpretation of the phenomenon, based on the circular inference framework. When the stimulus disappears, the belief converges to an attractor. The behavior of the system depends on the number and the value of the fixed points (here: $w_S = 1$; $a_P = 1.2$; $r_{on} = r_{off} = 1$ (symmetrical case) or $r_{on} = 1$; $r_{off} = 0.9$ (asymmetrical case)). **(B.,C.,F.,G.) No loops**—If there are no (descending) loops, when the stimulus disappears the beliefs converge to the prior **((B.) No implicit preference; (F.) Implicit preference)**. Consequently, for longer OFF-durations, the 2 survival probabilities (blue and red solid lines) either converge to 0.5 **((C.) No implicit preference)** or to symmetrical values **((G.) Implicit preference)**. In both cases, the stimulus is not stabilized for longer intervals. Interestingly, it is more stable compared to a continuous presentation (dashed lines). **(D.,E.,H.,I.) Descending loops**–Descending loops generate a bistable attractor **((D.) No implicit preference (H.) Implicit preference)**. Crucially, when they are strong enough, they cause stabilization for longer intervals **((E.) No implicit preference (I.) Implicit preference)**. Furthermore, in the biased case, survival probabilities converge to asymmetrical values.

and 6F). For the unbiased system, the model predicts that both survival probabilities (SP) will decrease toward 0.5 (chance) with a time constant that depends on the transition rates (Fig 6C). An SP in a biased system would reach symmetrical points above and below chance, with the values depending on the strength of the bias (Fig 6G). The longer the OFF-duration, the less temporal dependency there would be between subsequent percepts. Thus, without descending loops, there could not be any stabilization of the percept by an intermittent presentation for long "OFF" durations. For comparison, SP is shown for the continuous case (stimulation is not interrupted; in which case, we measure the survival probability in constant intervals; dashed lines).

The descending loops ($a_P > 0$) change the behavior of the system. The phase portrait of this system is presented in Fig 6D and 6H. Instead of one single point where all the trajectories meet, now we observe 2 clearly distinct basins of attraction, symmetrical for an unbiased system and asymmetrical for a biased system. As a result, the temporal stability of the percept is drastically increased, especially for long "OFF" durations (Fig 6E). In biased systems, the level of stabilization depends on whether we consider the dominant or nondominant percept. The probability of persistence of the dominant percept (if biased) always converges to a higher probability than the nondominant percept. In the example shown in Fig 6I, only the dominant stimulus is stabilized by intermittent presentation, while the nondominant percept SP converges to a chance level. In other cases, both the dominant and nondominant percept can be stabilized. The stabilization of both percepts increases with the level of descending loops and decreases with sensory gain, as shown in the next section.

An important comment needs to be made. The current version of the model does not predict a destabilization occurring for small OFF-durations, usually for values below 500 ms, as reported in some studies [55]. Other models have attributed this observation to short-term sensory adaptation [9]. To keep the model as simple as possible, we did not introduce sensory adaptation. However, such a short-term effect, occurring only at the time of stimulus presentation, would not affect the stabilization for long OFF-durations as predicted by the model with descending loops.

To summarize, dCI predicts the stabilization of bistable perception for longer OFF-periods. In addition, it makes specific predictions about the persistence of each interpretation separately, which could help to experimentally validate (or invalidate) this model.

## Bistable perception as a tool for investigating mental illness

So far, we have described a functional model of bistable perception, based on the notion of CI. Accumulating evidence supports the idea that circularity (and especially a small amount of descending loops) is a common property of the human brain, reflecting some inherent limitations of neural circuits [25,26]. However, it has also been suggested that CI could be the cause of several cognitive and/or perceptual disorders, including schizophrenia [22,24]. In a previous study, Jardri et al found that on average, patients with schizophrenia have stronger ascending loops compared to a group of matched healthy controls [25]. Additionally, it was evidenced that "positive" (i.e., psychotic) symptoms, including hallucinations and delusions, correlate with the amount of ascending loops (i.e., sensory evidence amplification), "negative" symptoms, including lack of motivation and anhedonia, correlate with the amount of descending loops (i.e., prior amplification), and finally, cognitive disorganization correlates with the total amount of loops ($a_S + a_P$). Considering these previous findings, an interesting question is what does the current dCI model predict the behavior of schizophrenia patients exposed to bistable stimuli?

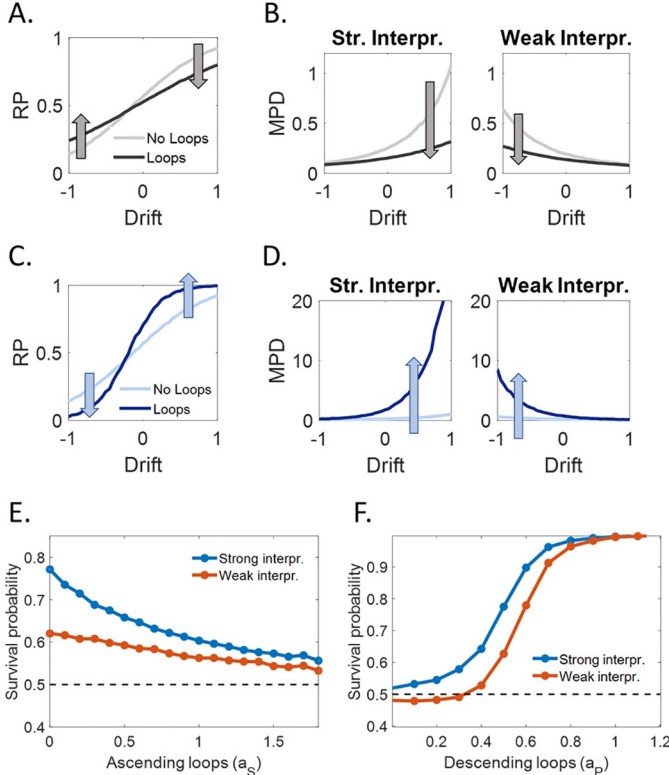

**Fig 7. Predicted effects of CI strength on bistable perception. (A.)** Relative predominance (RP) as a function of the strength of sensory evidence in favor (positive drift) or against (negative drift) the preferred configuration (i.e., $\mu_{noise}$) for increasing sensory gain (including ascending loops), from light to dark gray. **(B.)** Mean phase duration of the preferred and nonpreferred configuration. **(C.)** The same as (A.) but with no ascending loops and increasing descending loops, from light to dark blue. **(D.)** The same as (B.), with no ascending loops and increasing descending loops. **(E.)** The probability of persistence of the preferred (blue) and nonpreferred (red) configuration during the intermittent presentation of an ambiguous stimulus (stimulus duration 200 ms, OFF-duration 5 s) as a function of the ascending loops $a_S$ ($a_P = 0.5$). **(F.)** The same as (E.), but as a function of the descending loops $a_P$ ($a_P = 0$). All the other parameters were kept constant across simulations: $w_S = 1$; $r_{on} = 0.5$; $r_{off} = 0.48$.

Fig 7A and 7B illustrates the effect of ascending loops on the bias (relative predominance) and stability (mean phase duration). As previously shown, ascending loops increase the gain of the noise, facilitating the jumps between the 2 attractors. Consequently, our model predicts that patients with more severe hallucinations and delusions should be less biased in their responses (both due to inherent priors and visual cues) but also less stable (especially the interpretation that is supported by the visual cue). Specifically, the effect of ascending loops on relative predominance, although it might seem counterintuitive (over-counting of sensory evidence leads to a smaller effect of that evidence), illustrates the detrimental effect of the higher gain of noise on the accumulation of evidence.

In contrast, descending loops deepen the wells of the energy landscape and consequently, they produce the exact opposite effects. As shown in Fig 7C and 7D, the prediction would be that they increase both the bias and the stability of schizophrenia patients with more severe negative symptoms.

Similar stabilization and destabilization effects as a function of the level of ascending and descending loops are predicted for intermittent presentation (Fig 7E and 7F). In particular, increasing ascending loops (and thus, the sensory gain), leads to destabilization of both the dominant and nondominant percept (more precisely, both SP get closer to 0.5; Fig 7E). This

effect is in agreement with recent experimental results on schizophrenia patients [56,57]. In contrast, increasing descending loops stabilizes first the dominant percept, and then both the dominant and nondominant percepts (Fig 7F).

Finally, note that these predictions are not only qualitative but also quantitative. The results in Fig 7, as well as the shape of the stabilization curves in Fig 6, depend on 4 free parameters, the transition rates, overall descending loop strength $a$ and sensory gain $w_{int}$, all specifically related to generic parameters of perceptions applicable to many behavioral tasks. This could provide a foundation for parametric study of natural variation in the general population and psychiatric disorders, generalization over the results of different experiments (e.g., probabilistic decision tasks versus bistable perception), and raise the possibility of finding specific neural correlates of these variations (e.g., levels of E/I balance, effective connectivity between high-level and low-level areas, etc.) (see S4 Text).

## Discussion

In the present paper, we demonstrated that bistable perception could arise in a perceptual system where feedback based on the current beliefs corrupts the sensory inputs. In this scenario, expectations are reverberated back up and considered several times (forming descending information-loops), suboptimally amplifying prior beliefs and causing the system to «see what it expects» [24]. The emerging dynamical system can explain various intriguing features of bistable perception, including its mere existence. It artificially inflates the accumulated noisy information, leading to a system that perceives clearly, persistently and in alternation the two potential interpretations, with high levels of conviction. Such a dCI model is compatible with Levelt's laws and accounts for the stabilization of the percepts when the stimulus is presented intermittently.

Importantly, this model allowed us to make new predictions regarding bistable perception in physiological and pathological conditions. Each free parameter has a clear interpretation in terms of perceptual inference, can be directly estimated from behavioral data (see S4 Text), and can be generalized to predict behavior in other tasks (e.g., probabilistic decisions). Crucially, although descending loops could be necessary for bistability, they are not sufficient. Bistable stimuli need to lack crucial information that would clearly disambiguate them in a natural setting (such as depth cues). The perceptual system should expect the input distribution to differ between the two interpretations (otherwise they would be uninformative and disregarded) even if this is not the case for artificial stimuli used in bistable experiments (Fig 1B). Of note, completely ambiguous stimuli are, in fact, very rare [3,58] and unlikely to be learned from experience.

From the point of view of the underlying dynamics of perception, descending loops have important consequences beyond bistability. Due to their inherently stabilizing effect, a perceptual system can switch from a pure Bayesian integrator to a bistable attractor. By changing just the strength of descending loops, the perceptual system can transit between two decision-making strategies: Integration to bound [59,60] and attractor dynamics [61,62].

Beyond our model, various other implementations have been proposed to account for the unique characteristics of bistable perception. Mechanistic models have either focused on neural mechanisms [7,8,10] and/or on more abstract dynamical systems [6,9,12]. Nevertheless, those models are usually designed on an ad hoc basis and remain largely descriptive. With few exceptions (e.g., [45]), they are agnostic regarding the functional implication of bistability for perception and decision in general. In other words, although they may address the «what» questions (mechanisms and implementations), they are not addressing the «why» questions (epistemological questions).

To answer the second type of question, other groups have proposed functional models of bistable perception that approach the problem in a top-down fashion [17–21,63,64]. Like ours, those approaches focus on the type of problems that perceptual systems usually encounter (e.g., deal with uncertainty) and impose functional limitations (e.g., Markovian statistics, approximate Bayesian inference [65]). However, some of these models are abstract and do not specify neural mechanisms. Others are more complex and contain large numbers of free parameters, rendering them difficult to (in)validate experimentally.

In particular, an interesting model that bears some similarity with the dCI model was described by Hohwy and Friston [17] and formalized by Weilnhammer and colleagues [18]. Like dCI, it relies on a message passing algorithm, but instead of belief propagation, it is largely based on a simplified version of predictive coding [28,66,67]—predictive coding postulates that priors explain away sensory inputs while residual prediction error signals are fed-forward to higher regions to update beliefs. Importantly, top-down effects play a crucial role in both explanations of bistability. Instead of adding (descending) loops, the predictive coding model suggests that perception is biased by a stabilization prior, which depends on the current interpretation. This prior is constantly weakened by prediction errors emerging from evidence for the suppressed percept, via an exponential decay mechanism. A switch occurs when the evidence for the suppressed percept surpasses that for the dominant percept. Despite their similarities, the two models are not identical. While dCI is derived from first principles (inference in a hidden markov model, corrupted by loops), the predictive coding model relies on a number of ad-hoc assumptions, that nuance its normative character. For example, the precision of the stabilization prior is renormalized after each switch, resulting in strong and stable percepts; this is an important assumption, yet it's difficult to interpret it from a normative perspective.

Furthermore, several models were based on the idea that inference is approximated by a sampling process, without explicit calculation and knowledge of the exact posterior distribution [19–21]. In that case, bistable perception occurs because the perceptual system is assumed to take only one sample at each time step, resulting in high temporal correlations between samples. This is, in fact, a nuisance in this kind of algorithm, predicting a highly suboptimal form of perceptual inference (e.g., it takes a very long time to infer the exact probability distribution, and the corresponding estimates are much more variable than a maximum-a-posteriori estimate). Because of this limitation, perceptual inference by sampling might be far less performant than belief propagation (even with loops), raising the question of why our perceptual system would choose such a strategy. Additionally, it remains unclear whether those models could account for less trivial experimental results, including stabilization under an intermittent presentation.

Note that in our case, bistable perception could also be seen as a suboptimality resulting from descending loops (i.e., the estimated probability are not the correct ones given the real sensory evidence and prior knowledge). However, we predict that it mostly affects perception in rather unusual cases, e.g., for a fixed level of descending loops, stimuli that are both expected to be very reliable (high $w_S$) and in reality are highly ambiguous ($\mu_{noise}$ close to zero). Consequently, this unusual stimulus does not fit our generative model [68]. The effects could be far more subtle otherwise. In agreement with this hypothesis, we found that CI only rarely affects choices in randomly selected probabilistic inference problems (i.e., random graphs, see [22]).

The dCI model presented in this paper is normative (i.e. derived from first principles; strictly speaking, normativity is violated due to the loops) but can also be seen as descriptive due to its closed-form solution. Switches in perceptual bistability are driven by noise in agreement with existing evidence [13–15]. In contrast to models based on lateral inhibition between local populations, bistable perception is interpreted as a brain-wide phenomenon linked to inhibitory control of feedforward and feedback processes (as is generally required

for hierarchical perceptual inference [22]). Its dynamical behavior has important similarities with that of other attractor models [12], but the bistable attractor is hereby not imposed to explain certain features of bistability, but instead a direct consequence of the descending loops. In the same vein, our model makes a clear distinction between a bias induced by sensory evidence and bias resulting from the system's implicit preference (prior knowledge), thus enabling the generation of asymmetries in the absence of stimulation (intermittent presentation).

Another important feature of bistable perception, shared by human and nonhuman observers, is the distribution of dominance durations. Although there is considerable variability in the mean phase duration between participants (but also within participants and between conditions or stimuli), there is an impressive similarity in the shape of the distribution of phase durations, relatively well approximated by a gamma or log-normal distribution [69–71] (but see also [72]). The dCI model, like all the noise-driven attractor models, generates exponential distributions of phase durations [12]. Several extensions of the model can engender gamma-like distributions, in which simple mechanisms are added on an ad-hoc basis. For example, one could assume that inference is preceded by filtering, which takes place at the very first levels of the sensory hierarchy (e.g. retina, LGN in case of visual inputs); filtered noise is smoother than gaussian noise and precludes the occurrence of fast switches. Alternatively, one could introduce an adaptation-like mechanism (see also [12]); in the dCI context, this could be implemented as time-dependent transition rates, e.g. as a form of learning. Finally, a third option is to replace MAP with a more complex decision criterion, e.g. a more conservative criterion, implemented as a moving threshold, where switches occur only when there is substantial evidence in favour of the opposite interpretation.

It has been argued that CI are linked at the neurophysiological level to an imbalance between neural excitation and inhibition in favor of excitation [24,27]. This imbalance might concern only local microcircuits, encompassing pyramidal cells and local interneurons (Fig 1D), or more global networks, potentially involving thalamocortical or corticostriatal long-range connections [24]. Although both are plausible implementations of loops, local interneurons make a better candidate in the particular case of bistable perception. Indeed, it has been argued that bistability is a rather low-level process mainly occurring within the visual cortex ([4,73,74]; but see [75,76], arguing for the involvement of high-level areas) while the involvement of local inhibition is also supported by pharmacological evidence [77].

Apart from normal brain functioning, CI has been used to account for clinical dimensions in schizophrenia [22,25]. Our model implies that generic mechanisms involved in hallucinations and delusions could also explain common perceptual phenomena, such as bistable perception, in agreement with the idea that psychosis may exist along a continuum with normal experience [78–81]. Nevertheless, when and how exactly those mechanisms go awry and generate pathological symptoms remains an open question. In addition, the present model provides a dynamical system interpretation of CI models, relating them to other influential frameworks [82–84].

Could circularity offer a relative advantage to perceptual systems or is it simply a manifestation of the inherent limitations of neural systems? Our present results suggest that a system performing exact inference with ambiguous information could be more vulnerable to noise and have difficulties in forming stable percepts. Moderate descending loops could improve the system, allowing rapid and robust decisions even when evidence is not conclusive (after all, both "fighting" and "fleeing" are better than standing still; a similar explanation was suggested by Moreno-Bote and colleagues, who interpreted bistability as exploratory behavior under uncertainty [45]). Moving a step further, a system with flexible descending loops (e.g., a system that can regulate its E/I balance through neuromodulators, such as dopamine, serotonin or

acetylcholine [85,86]) could vary the perceptual strategy from impulsive to deliberative in accordance with task requirements. This suggestion, although speculative, could reconcile the present results with evidence showing a balance between excitation and inhibition at different scales [87–89] and is furthermore easily testable (e.g., by measuring E/I balance during bistability and during stimulation with unambiguous stimuli).

In conclusion, we described bistable perception as a probabilistic inference process, under the influence of amplified priors due to the presence of descending loops in the cortical hierarchy. The model explains why bistable perception occurs in the first place and qualitatively predicts several of its properties. Additionally, it has important implications for the neural correlates of bistability and the relation between normal brain functioning and pathology, ultimately linking computation, behavior and neural implementation.

## Supporting information

**S1 Text. Mathematical derivations.**
(DOCX)

**S2 Text. Bifurcation analysis.**
(DOCX)

**S3 Text. Phenomenology of bistable perception.**
(DOCX)

**S4 Text. Parameter recovery.**
(DOCX)

## Author Contributions

**Conceptualization:** Pantelis Leptourgos, Vincent Bouttier, Renaud Jardri, Sophie Denève.

**Data curation:** Pantelis Leptourgos, Vincent Bouttier, Renaud Jardri, Sophie Denève.

**Formal analysis:** Pantelis Leptourgos, Vincent Bouttier, Renaud Jardri, Sophie Denève.

**Funding acquisition:** Pantelis Leptourgos, Renaud Jardri, Sophie Denève.

**Investigation:** Pantelis Leptourgos, Vincent Bouttier, Renaud Jardri, Sophie Denève.

**Methodology:** Pantelis Leptourgos, Vincent Bouttier, Renaud Jardri, Sophie Denève.

**Project administration:** Renaud Jardri, Sophie Denève.

**Resources:** Pantelis Leptourgos, Vincent Bouttier, Renaud Jardri, Sophie Denève.

**Software:** Pantelis Leptourgos, Vincent Bouttier, Renaud Jardri, Sophie Denève.

**Supervision:** Renaud Jardri, Sophie Denève.

**Validation:** Pantelis Leptourgos, Vincent Bouttier, Renaud Jardri, Sophie Denève.

**Visualization:** Pantelis Leptourgos, Vincent Bouttier, Renaud Jardri, Sophie Denève.

**Writing – original draft:** Pantelis Leptourgos, Vincent Bouttier, Renaud Jardri, Sophie Denève.

**Writing – review & editing:** Pantelis Leptourgos, Vincent Bouttier, Renaud Jardri, Sophie Denève.

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
