## [Decision Letter · Decision Letter 0]

17 Apr 2020

Dear Dr Leptourgos,

Thank you very much for submitting your manuscript "A functional theory of bistable perception based on dynamical circular inference" for consideration at PLOS Computational Biology.

As with all papers reviewed by the journal, your manuscript was reviewed by members of the editorial board and by several independent reviewers. In light of the reviews (below this email), we would like to invite the resubmission of a significantly-revised version that takes into account the reviewers' comments.

From the reviewer comments it should be pretty clear that there are just a few, but serious concerns that need to be addressed. We would call your attention to reviewer 1's comment about model fitting, and especially reviewer 2's comments about Levelt’s laws. If you believe that you can address all the reviewers points we would be happy to reconsider the manuscript. We would like to emphasize that the issues raised by the reviewers concerning Levelt's laws possibly indicate fundamental problems with the modeling approach, so a revised manuscript might be rejected without being sent for re-review if we feel that these problems have not been adequately addressed (though we sincerely hope they can be addressed).

We cannot make any decision about publication until we have seen the revised manuscript and your response to the reviewers' comments. Your revised manuscript is also likely to be sent to reviewers for further evaluation.

Sincerely,

Ulrik R. Beierholm

Associate Editor

PLOS Computational Biology

Samuel Gershman

Deputy Editor

PLOS Computational Biology

Reviewer's Responses to Questions

**Comments to the Authors:**

Reviewer #1: The authors present a new computational approach toward bistable perception. Based on the established algorithm of circular inference, they deduce minimal conditions under which bistable perception can occur in this framework. With this, the authors place bistable perception in the general context of perceptual inference. Lastly, they relate their model to alternations in perceptual inference related with psychotic symptoms.

Next to analytical methods, the authors performed simulation analyses and compared the model predictions to a number of empirical characteristics of bistable perception (Levelt’s laws and the stabilization of bistable perception by intermittent presentation of ambiguous stimuli).

I think that the circular-inference approach to bistable perception outlined in the manuscript is highly relevant for two reasons: Firstly, it deduces the perceptual dynamics of bistable perception from general considerations of perceptual inference. To my mind, such a functional take on bistability is important, since it may help translate perceptual phenomena observed largely in a laboratory context (eg, the frequent perceptual transitions experienced during perceptions) to the characteristics of perception in real-world scenarios. Secondly, by inverting the circular inference model based on behavior, researchers may be able to quantify the relative contribution sensory evidence and prior knowledge on perceptual inference. This may proffer new opportunities in the study of alterations in perceptual inference (eg, positive psychotic symptoms in schizophrenia).

Although I am very sympathetic toward this work, I have several concerns and wishes for clarification:

General Comments

First, broadly, the authors’ claim that this model can be fitted to behavioral data (and thus be useful to study perceptual function in health and disease) does not seem to be backed up empirically in the manuscript. Personally, I would think that the authors could strengthen their circular-inference model of bistable perception significantly if they could show that the latent variables of the model (eg, weights for ascending/descending loops, feed-forward weight ws, the response variable theta, rates of change ron/roff) can be reliably estimated from data. To my mind, this would mean that, when simulating behavioral responses for a set of latent variables, such latent variables could be recovered from the simulated data.

Second, I have some specific wishes for further clarification regarding the methods and analysis, which I outline in detail below. Lastly, I have a few recommendation on how to improve the connection of the authors’ findings to the existing literature. Specific comments are below.

Major comments:

1. I did not fully understand the role of the decision threshold theta. To my understanding, in this circular-inference (CI) model, setting theta to a sufficiently high value should be necessary to maintain stable perceptual states (view from above; view from below in the example of the Necker Cube). For low values of theta, I could imagine that spontaneous fluctuation in L should lead to frequent switches between dominant perceptual states. Yet, in the method section, the authors note that:

“(…) in the case of continuous presentation, it is necessary to set to a sufficiently high positive value to obtain robust perceptual switches. If = 0, the percept would switch multiple times (just because of the noisy input causing small random fluctuations in ) around the time of perceptual transitions.”

Specifically, I did not understand why the stabilizing effect of a high decision threshold stabilized perception only around the time of perceptual transitions,

2. On a more general view, I would find it helpful to see an illustration of the effects of theta on the model’s predictions.

“In our model, a nonzero decision threshold precludes percepts with very short durations. As a result, the distribution resembles a gamma or log-normal distribution. The rising phase and peak of this distribution depend on the decision threshold, but the tail of the distribution does not and is imposed by the mathematical properties of bistable dynamics driven by noise.”

Did I understand correctly that, in the presence of descending loops, it is only due to the decision threshold that simulated phase durations are distributed in a gamma/log-normal distribution? Is the location of the peak of the distribution uniquely determined by the value of the decision threshold? Is there any relation between the energy landscapes shown in Figure 3 and the theta parameter? Moreover, is the minimum value of the decision threshold that is necessary to induce stable perceptual inference correlated with the standard deviation of sensory evidence / the likelihood function?

3. In a related point, I would need additional clarifications on how the role of theta relates to the function of descending loops:

“Note that large values of can lead to a distribution of phase duration similar to the system of descending loops. However, while the distribution of phase duration cannot be considered proof of the presence of circular inference, the resulting confidence is often below the decision thresholds. This may preclude the emergence of strong and stable percepts in the absence of descending loops.”

Here, the authors introduce the concept of “confidence”. If I understood it correctly, high-confidence perceptual states only emerge in the presence of descending loops. I would find it helpful if the authors could contextualize this to the existing literature on bistable perception:

A number of studies has devoted a lot of attention of mixed percepts during bistability (eg., Knapen 2011). Do the authors assume that such mixed percepts (low-confidence/high-uncertainty perceptual states) arise at the time of state transitions between the energy wells in Figure 3c? How would the energy landscape look like for other types of bistable stimuli that show sudden transitions, such as discontinuous structure-from-motion stimuli (eg., Weilnhammer 2013)?

4. With regard to perceptual biases: From Figure 3d, should it be concluded that the CI model assumes a difference in confidence when there is a bias between perceptual alternatives? In the example of the Necker Cube, this would mean that the view-from-below is generally associated with reduced confidence. To my mind, this would be an important prediction of the CI model. Are the authors aware of any empirical evidence for this model prediction?

5. In the section on Levelt’s 4th law, the authors investigate the effect on an increase in the strength of both interpretations on the alternation rate of a bistable stimulus. They captured this increase in stimulus strength by increasing the variance of the noise distribution. This choice did not seem straightforward to me. Several alternatives would also seem plausible to me: Could an increase in stimulus strength be reflected by a decrease in variance of the noise distribution? Or by a modulation in variance of the likelihood distribution?

Minor comments

With regard to the Abstract, I would like to make a few suggestions that could render the content more accessible to the naïve reader:

1. While these points become clear after reading the manuscript, my personal impression was that they are difficult to understand on the basis of the abstract and general knowledge about bistable perception. Readers without a background in computational modelling of bistable perception might have a hard time understanding these points.

“We show that in the face of ambiguous sensory stimuli, circular inference can

turn what should be a leaky integrator into a bistable attractor switching between two highly trusted interpretations. (…) Since it is related to the generic perceptual inference mechanism, this approach can be used to predict the tendency of individuals to form aberrant beliefs

from their bistable perception behavior.”

2. Maybe I have overlooked something, but while the main text contains a section of psychotic symptoms in schizophrenia patients, I could not find a discussion on cognitive functions in non-clinical populations.

“Overall, we suggest that feedforward/feedback information loops in hierarchical neural networks, a phenomenon that could lead psychotic symptoms when overly strong, could also underlie cognitive functions in nonclinical populations.”

With regard to the introduction, I have a few additional comments:

3. If I understood correctly, the authors introduced perceptual inference during bistable perception as “suboptimal”:

“In most cases, this task is performed very accurately, and the correct interpretation is found. Sometimes, perceptual systems fail to detect any meaningful interpretation (e.g., when sensory evidence is too degraded) or converge to the wrong interpretation. Finally, a third possibility occurs (mainly in lab conditions [3]) when ambiguity is high; the system detects more than one plausible interpretations but instead of committing to one interpretation, it switches every few seconds, a phenomenon known as bistable perception [4].

(….) Crucially, there is a discrepancy between the real input and the input assumed by the internal model. This, together with the loops, predicts the suboptimal inference at the heart of

bistable perception (Figure 1; caption).

I was wondering whether the authors could add a little more detail as to why they view perceptual inference is suboptimal or incorrect. As they authors note throughout the manuscript, truly ambiguous images (eg. the line drawings of a Necker cube, disparate monocular inputs in case of binocular rivalry) are very rare.

Could it also be that, because of the extremely low probability of a fully ambiguous real-word cause of sensory input, committing to one highly trusted stimulus interpretation is indeed adaptive/optimal? This thought also appears in the discussion (“Moderate descending loops could improve the system, allowing rapid and robust decisions even when evidence is not conclusive”)

Reviewer #2: General comment

In this paper, Pantelis Leptourgos and colleagues propose 'circular inference', a model of hierarchical Bayesian inference, for the modeling of bistable perception. They suggest that bistability can be conceived as a process whereby sensory responses trigger activity in higher-level areas but are also modulated by feedback projections from these same areas, thus reverberating prior beliefs in the cortical hierarchy. They show that in the face of ambiguous sensory stimuli, circular inference can turn what should be a leaky integrator into a bistable attractor that switches between two alternative interpretations. Moreover, they report that their model is able to capture various aspects of bistable perception, including Levelt’s laws and the stabilizing effect of intermittent stimulus presentation. They speculate that feedforward/feedback information loops in hierarchical neural networks, a phenomenon that could lead to psychotic symptoms when overly strong, could also underlie cognitive functions in nonclinical populations.

Overall, this is a very interesting, timely and well-written paper that adds an interesting new approach to the modeling of bistable perception. Since this algorithmic model is also based on established theories of cortical neural circuit function, such as E/I balance it is a promising candidate to link modeling at the algorithm level to the level of neural implementation. In addition, it may be well suited to the modeling of altered perceptual inference in neuropsychiatric disorders such as schizophrenia, as the authors point out. I have a few concerns and questions that I would like to ask the authors to consider. In particular, there seem to be a few problems with the part regarding Levelt's laws that need to be addressed (see below for details).

Specific points

1. In the introduction, the authors state that a few important questions have remained unanswered by previous modeling approaches. The first is, why a system should form "such strong percepts based on unreliable sensory evidence". I'm not sure that the sensory evidence that typically gives rise to bistable perception is what I would call "unreliable". It is perceptually ambiguous, which means that it is compatible with two competing perceptual interpretations. However, as opposed to noisy stimuli, the evidence is quite clear-cut and reliable. It's not the sensory information that's unreliable, it's the perceptual system that has a hard time "making up it's mind" which of two clear-cut interpretations to go for. It is therefore not obvious to me what should be puzzling about the fact that two "strong" percepts are formed.

The second question is why percepts persist "instead of switching rapidly". To my mind, this question has been answered quite satisfactorily, from a normative perspective, by Hohwy et al. 2008. In their conceptual model (which was later implemented as a computational model, see Weilnhammer et al. 2017), it is the current percept that determines the prediction of what the sensory input is caused by. This prediction is fed back from higher to lower hierarchical levels and thereby stabilizes perception. This is very similar to what the authors suggest in their current work, so I'm struggling to see what is new here (again, from a normative perspective).

The third question is "how the behavior of individuals in bistable perception tasks may predict their performance in other probabilistic inference tasks". This is a potentially interesting topic, but I did not find it addressed in this paper.

2. Related to the second of the above questions: From a normative perspective and conceptually, the circular inference model of bistable perception is extremely similar to the predictive coding model proposed by Hohwy et al./Weilnhammer et al.. Both models assume top-down projections (feedback/descending loops) to stabilize and bottom-up projections (feedforward/ascending loops) to destabilize perception. Since these two models are so closely related, I think it would be worthwhile to add a paragraph to the discussion section where the circular inference model is discussed in the light of this previous model, how the new model differs from the predictive coding model, and how it may go beyond it?

3. Why is the theta parameter needed at all, as it only has a thresholding function to avoid fast reversals. It is not needed in the presence of descending loops, and is stated therefore not to be considered further (Lines 352 f.)? So why is it needed in the model at all? And what could be an equivalent of this parameter in terms of a neural mechanism or function? Please clarify.

4. Maybe I din't get the point here, but I'm not sure I understood why sensory evidence is referred to as "noise" (e.g. line 418). The sensory data do contain some information, not only noise, don't they?

5. Levelt's 2nd law is misrepresented and it therefore seems that the authors' conclusions in this regard are not tenable. It is stated that manipulating the strength of one perceptual interpretation "mainly affects the persistence of the stronger interpretation". This is not what Levelt's 2nd law says. In fact it states that increasing stimulus strength for one eye will NOT affect the average perceptual dominance duration of that eye’s stimulus. Instead, it will reduce the average perceptual dominance duration of the other eye’s stimulus. What results from the model presented here is exactly the opposite.

6. Levelt's 3rd law is similarly misrepresented. The authors state "… that increasing the difference in the stimulus strength between the two perceptual interpretations should result in a decrease in the perceptual alternation rate". Again this is not what Levelt's 3rd says. What it does state is that increasing the strength of one stimulus will increase the alternation rate. Increasing the strength of one stimulus effectively increases the difference in stimulus strength, therefore the authors' statement says exactly the opposite and, again, what results from the model does not conform to Levelt's 3rd law.

7. The model manipulation that is used to test Levelt's 4th law is inconsistent with what was done to test the other three laws. For the first three laws, which entail a difference in stimulus strength, stimulus differences are modeled by changing the noise drift parameter. This is obviously not possible if the strength of both stimuli needs to be changed in the same direction. Instead, the authors resort to another parameter, the noise distribution. In other words, this looks like the authors model changes in stimulus strength by changing ad libitum whichever model parameter suits them best. There may be a good reason to do it this way, but if there is, it should be explained.

8. Line 324: "resulting in an exponential decay observed in the distribution of dominance durations". It would be very helpful if the dominance distributions were presented in a figure.

9. Line 127: It is misleading to say that the Necker cube is "equally compatible with 2 different 3D cubes", as it is actually more compatible with the view from above (as the authors also state further down in the paper).

10. Line 348: "see the Results section". This IS the results section.

11. "Severity of loops" (used multiple times) should probably better read "Strength of loops".

12. Line 714: "flying" should read "fleeing".

Signed: Philipp Sterzer

**Have all data underlying the figures and results presented in the manuscript been provided?**

Reviewer #1: Yes

Reviewer #2: Yes

PLOS authors have the option to publish the peer review history of their article (what does this mean?). If published, this will include your full peer review and any attached files.

Reviewer #1: Yes: Veith Weilnhammer

Reviewer #2: Yes: Philipp Sterzer
---

## [Decision Letter · Decision Letter 1]

30 Oct 2020

Dear Dr Leptourgos,

We are pleased to inform you that your manuscript 'A functional theory of bistable perception based on dynamical circular inference' has been provisionally accepted for publication in PLOS Computational Biology.

Best regards,

Ulrik R. Beierholm

Associate Editor

PLOS Computational Biology

Samuel Gershman

Deputy Editor

PLOS Computational Biology

You will notice that reviewer 1 has some minor comments and recommendations that you should consider when submitting your final manuscript. However no response to the reviewer is required.

Reviewer's Responses to Questions

**Comments to the Authors:**

Reviewer #1: The review is uploaded as an attachment.

Reviewer #2: I thank the authors for the clarifications and the careful revision. I have no further comments.

**Have all data underlying the figures and results presented in the manuscript been provided?**

Reviewer #1: Yes

Reviewer #2: Yes

PLOS authors have the option to publish the peer review history of their article (what does this mean?). If published, this will include your full peer review and any attached files.

Reviewer #1: **Yes: **Veith Weilnhammer

Reviewer #2: No

---

## [Editor Report · Acceptance letter]

3 Dec 2020

PCOMPBIOL-D-20-00430R1 

A functional theory of bistable perception based on dynamical circular inference

Dear Dr Leptourgos,

I am pleased to inform you that your manuscript has been formally accepted for publication in PLOS Computational Biology. Your manuscript is now with our production department and you will be notified of the publication date in due course.

With kind regards,

Nicola Davies
